materials science

peanut protein isolate, arachin, conarachin, wood adhesives, urea, water resistance

**Author for correspondence:**
Fusheng Chen
e-mail: fushengc@haut.edu.cn

†These authors contributed equally to this work.

This article has been edited by the Royal Society of Chemistry, including the commissioning, peer review process and editorial aspects up to the point of acceptance.

# Effect of urea concentration on properties of peanut protein isolate, arachin and conarachin-based adhesives during urea-epichlorohydrin modification

Chen Chen[1,2,†], Yan Du[1,†] and Fusheng Chen[1]

[1]College of Food Science and Engineering, Henan University of Technology, 100, Lianhua Street, High-tech, Zhengzhou 450001, Henan, People's Republic of China
[2]College of Food Science and Nutritional Engineering, China Agricultural University, Beijing 100083, People's Republic of China

CC, 0000-0001-5914-228X; YD, 0000-0003-4123-8720; FC, 0000-0002-8201-1234

To lay a theoretical basis for the preparation of peanut protein-based adhesives and promote the sustainable development of the adhesive industry, properties of peanut protein isolate (PPI), arachin and conarachin-based adhesives modified by urea and epichlorohydrin (ECH) were investigated under different urea concentrations. When the urea concentration was 2 mol l$^{-1}$, the wet shear strength of the PPI-based adhesive was 1.24 MPa with the best water resistance. With the increase of urea concentration from 0 to 4 mol l$^{-1}$, the apparent viscosity of the PPI-based adhesive increased from 3.87 to 136.80 Pa s and the solid content increased from 18.11% to 31.11%. Compared with conarachin-based adhesive, the properties of arachin-based adhesive were improved more obviously during the combined modification. Scanning electron microscopy images illustrated that when the urea concentration was 2 mol l$^{-1}$, the surface of the PPI-based adhesive was more compact and smoother, which was beneficial to the improvement of water resistance and related to the structure changes of arachin and conarachin components. Fourier-transform infrared spectroscopy results indicated that different urea concentrations caused the change of ester and ether bonds in the PPI-based adhesive, which was mainly related to arachin component. Thermogravimetry results suggested that when the urea concentration was 2 mol l$^{-1}$, the decomposition temperature of protein skeleton in the PPI-based adhesive reached a maximum of 314°C exhibiting the highest thermal stability. The improvement of the thermal stability of conarachin was greater than that of arachin during the combined modification.

# 1. Introduction

Recently, synthetic formaldehyde adhesives, for instance, phenol-formaldehyde, urea-formaldehyde and melamine-formaldehyde are widely used in the wood synthesis industry due to their excellent bonding performance and low cost [1]. Nevertheless, the release of formaldehyde during the production and use of these synthetic formaldehyde adhesives will do great harm to human health and the environment [2]. Therefore, the development of environment-friendly biomass adhesives with excellent properties, not only has exceptional economic value, but also has considerable environmental and social significance [3].

Based on renewable materials, such as plant protein, starch and lignin, many attempts have been conducted to seek substitutes for synthetic formaldehyde adhesives [4]. Considering the supply of biomass resources and economic benefits, researchers are increasingly interested in plant protein-based adhesives [5,6]. However, the poor water resistance of natural plant protein-based adhesives is the main reason for limiting its wide application in industrial production [7]. Numerous studies have been carried out to enhance water resistance. Silane coupling agent KH-560 can be used in the modification of soy protein adhesive, and the water resistance of the obtained soya bean protein adhesive was significantly improved [8]. Liu *et al*. [9] found that undecylenic acid with hydrophobic carbon chain and active carboxyl group was an efficient modifier to enhance the wet shear strength of soya protein isolate-based adhesives. Wei *et al*. [10] discovered that after adding zinc oxide, the wet shear strengths of the sesame protein-based adhesives were improved due to the chelation reaction between sesame protein and $Zn^{2+}$.

China is the major country of peanut consumption and production [11,12]. As the main by-product during peanut oil pressing, peanut meal (PM) is rich in peanut protein and has the advantages of wide source, low price and renewable. However, due to the fact that the processing technology of PM is not much advanced, most PM cannot be fully used, and most of them were used as animal feed or fertilizer, resulting in a large amount of waste of peanut protein resources [13].

Our research group has previously reported that the combined modification of urea solution and epichlorohydrin (ECH) can improve the properties of PM-based adhesive, especially the water resistance, which makes peanut protein a potential raw material for adhesives [14]. Moreover, for the sake of further exploring the performance variations of peanut protein-based adhesives under different urea concentrations, and at the same time eliminate the influence of non-protein components in PM on the results as much as possible, peanut protein isolate (PPI) with protein content above 90% was employed as the material to prepare the PPI-based adhesive in this research.

In addition, two major components, arachin (14S) and conarachin (7.8S and 2S) account for more than 90% of PPI [15–17]. The polypeptide side chains of arachin and conarachin contain diverse active groups, such as hydroxyl (-OH), amino (-NH$_2$), carboxyl (-COOH) and thiol (-SH), which are instrumental in reacting with multifarious chemical modifiers. The structure and performance changes of the PPI-based adhesive after combined modification may be associated with its arachin and conarachin components. Consequently, arachin and conarachin-based adhesives were prepared under the same conditions of the PPI-based adhesive.

The main goal of this work was to explore the performance of the PPI-based adhesive under different urea concentration during the combined modification of urea and ECH and, simultaneously, to investigate the internal relationship of the structure and performance variations between the PPI-based adhesive and its arachin and conarachin components. The development of peanut protein-based adhesives can not only reduce formaldehyde pollution and provide ideas for the preparation of biomass wood adhesives, but also promote the efficient and high-quality utilization of peanut protein resources and enhance the added value of peanut products.

# 2. Material and methods

## 2.1. Materials

Defatted PM flour with 50.12% of peanut protein (80 meshes) was commercialized by Zhengyang Sannong Seed Industry Co., Ltd (Henan, China). The analytical grade sodium hydroxide (NaOH), hydrochloric acid (HCl), ammonium sulfate ((NH$_4$)$_2$SO$_4$), urea and ECH were supplied by Luoyang Reagent Co., Ltd (Henan). Poplar veneer (9%–11% moisture content, $300 \times 300 \times 2.0$ mm) was provided by Bio Biologic Co., Ltd (Henan).

## 2.2. Preparation of peanut protein isolate, arachin and conarachin

### 2.2.1. Preparation of peanut protein isolate

PPI was prepared in accordance with the previous method with a few modifications [18]. The PM flour and distilled water were evenly mixed at a ratio of 1 : 10 and adjusted to pH 9.0 with the NaOH solution (1 mol l$^{-1}$). After stirring at 50°C for 100 min, centrifuge at 4000$g$ for 20 min. Then adjust the pH of the supernatant obtained to 4.5 with the HCl solution (1 mol l$^{-1}$) and stir at 25°C for 30 min. After centrifuging at 4000$g$ for 20 min, the precipitate in the lower layer was washed to neutral, and then the PPI was obtained after freeze-drying.

### 2.2.2. Preparation of arachin and conarachin

Arachin and conarachin were prepared according to the method described by Chiou [19] with some modifications. The PM flour was mixed with phosphate buffer solution (pH 7.9) at a ratio of 1 : 20. After stirring at 25°C for 1 h, centrifuge at 4°C and 8000$g$ for 20 min. Then, a certain amount of $(NH_4)_2SO_4$ was added to the supernatant obtained to make the solution saturation reach 40% (electronic supplementary material). The slurry was stirred at 4°C for 2 h and centrifuged at 4°C and 8000$g$ for 20 min. The precipitate was redissolved with the above phosphate buffer solution (material to liquid ratio of 1 : 5), and the arachin was obtained after freeze-drying. Some $(NH_4)_2SO_4$ was introduced to the centrifugal supernatant to make its saturation reach 60%. After stirring at 4°C for 1 h, the slurry was centrifuged at 4°C and 8000$g$ for 20 min. Some $(NH_4)_2SO_4$ was introduced again to the obtained supernatant to achieve the solution saturation of 85%. The slurry was stirred at 4°C for 2 h and centrifuged at 4°C and 8000$g$ for 20 min. The obtained precipitate was redissolved with the above phosphate buffer solution (material to liquid ratio of 1 : 5), and the conarachin was collected after freeze-drying.

## 2.3. Preparation of peanut protein isolate, arachin and conarachin-based adhesives

### 2.3.1. Preparation of arachin-based adhesives

Twenty grams of PPI and 100 ml urea solution (0, 1, 2, 3 and 4 mol l$^{-1}$) were added into the reaction vessel (condensation tube, thermometer, magnetic rotor), respectively, and stirred at 50°C for 1 h. Then, 6 wt% ECH (according to the total adhesive mass) was introduced and the resulting slurry was stirred at 50°C for 1 h. After cooling, the modified PPI-based adhesives I0, I1, I2, I3 and I4 were prepared respectively.

### 2.3.2. Preparation of arachin-based adhesives

Twenty grams of arachin and 100 ml urea solution (0, 1, 2, 3 and 4 mol l$^{-1}$) were added into the reaction vessel (condensation tube, thermometer, magnetic rotor), respectively, and stirred at 50°C for 1 h. Then, 6 wt% ECH (according to the total adhesive mass) was introduced and the resulting slurry was stirred at 50°C for 1 h. After cooling, the modified arachin-based adhesives A0, A1, A2, A3 and A4 were prepared.

### 2.3.3. Preparation of conarachin-based adhesives

Twenty grams of conarachin and 100 ml urea solution (0, 1, 2, 3 and 4 mol l$^{-1}$) were added into the reaction vessel (condensation tube, thermometer, magnetic rotor), respectively, and stirred at 50°C for 1 h. Then, 6 wt% ECH (according to the total adhesive mass) was introduced and the resulting slurry was stirred at 50°C for 1 h. After cooling, the modified conarachin-based adhesives C0, C1, C2, C3 and C4 were prepared, respectively.

## 2.4. Preparation of the plywoods

Three-layer poplar veneers were interlaced vertically and horizontally, and the adhesive was applied on both sides of the middle poplar veneer. One-sided glue amount was 220$g$ m$^{-2}$. Then, the three-layer poplar veneer was hot-pressed (1.0 MPa, 130°C, 600 s) using an XLB-350 (Qi Cai Hydraulic Machinery Co., Ltd, Shanghai, China). The plywoods were laid for 24 h at room temperature after hot pressing and then cut into test-specimens (100 × 2.5 mm) as shown in figure 1.

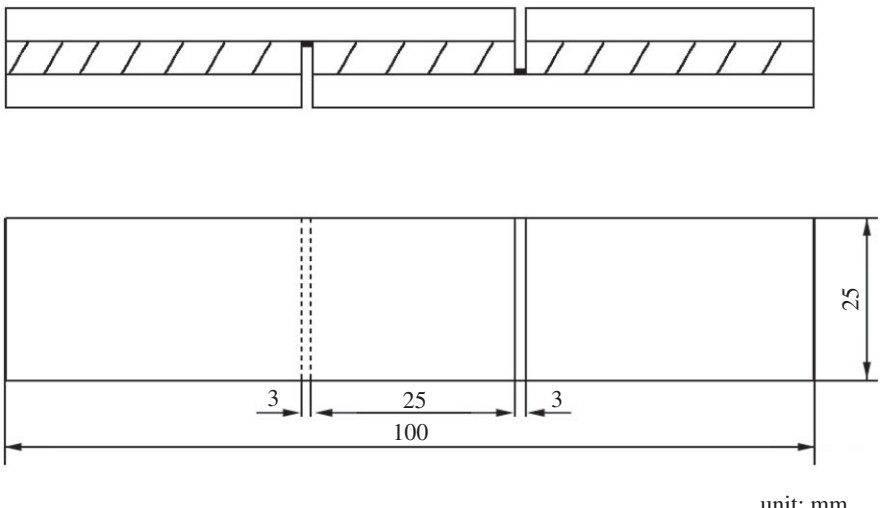

unit: mm

**Figure 1.** The shape and size of plywood test-specimens.

## 2.5. Wet shear strength measurements

On the basis of the requirements of class II plywood in the Chinese National Standards GB/T 9846-2015 and GB/T 17657-2013, the wet shear strength measurements were completed by the MWD-10 instrument (Si Da Te Test Co., Ltd, Shandong, China). Firstly, the test-specimens were pre-treated by soaking in hot water (63°C) for 3 h and then cooling to room temperature. The load was evenly applied with a tensile speed of 10 mm min$^{-1}$ until the glued surface was damaged. The peak load values of 12 parallel test-specimens in each group were recorded to obtain the average value. The shear strength was calculated from equation (2.1).

$$\text{Shear strength (MPa)} = \frac{\text{Force (N)}}{\text{Gluing area (m}^2)}. \tag{2.1}$$

## 2.6. Apparent viscosity measurements

About 1 ml of the adhesive was placed on the disc of the RS-6000 rheometer (Thermo Fisher Scientific Corporation, Waltham, MA). The apparent viscosity was tested at the stable shear condition (rotor model: pp35ti, shear rate: 1 s$^{-1}$, shear time: 1 min, temperature: 25°C). Each group was repeated three times to obtain the average value.

## 2.7. Solid content measurements

About 3$g$ of the adhesive (weight $\alpha$) was dried to constant weight at 105°C (weight $\beta$). The solid content was measured according to equation (2.2) ($n = 3$).

$$\text{solid content (\%)} = \frac{\beta}{\alpha} \times 100\%. \tag{2.2}$$

## 2.8. Characterization

The modified adhesive samples were dried to constant weight at $120 \pm 2°C$ and characterized as described previously [14]. In brief, protein block size and the surface structure of the samples were observed by the Quanta Feg 250 (FEI Company, USA) scanning electron microscope (SEM). Fourier-transform infrared (FTIR) spectroscopy was measured from 400 to 4000 cm$^{-1}$ at a resolution of 4 cm$^{-1}$ and 32 scans by a Nicolet 6700 spectrometer (Thermo Nicolet Corporation, Madison). Thermogravimetry (TG) analysis was tested by a TA Q50 machine (Waters Company, USA), and samples were heated from 25 to 600°C at a heating rate of 10°C min$^{-1}$ under nitrogen gas.

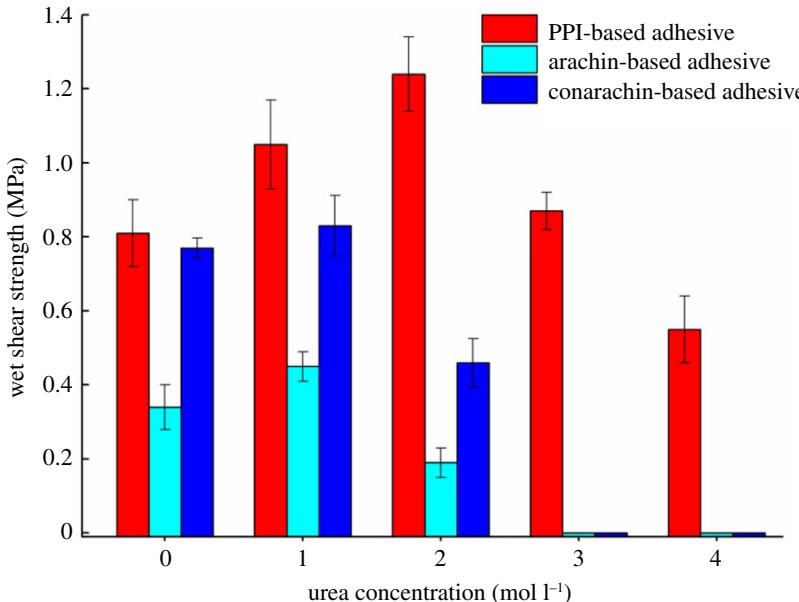

**Figure 2.** Wet shear strength of the PPI, arachin and conarachin-based adhesives.

# 3. Results and discussion

## 3.1. Wet shear strength of the peanut protein isolate, arachin and conarachin-based adhesives

Water resistance is a pivotal characteristic that determines the durability of adhesive bonding. The level of the wet shear strength reflects the quality of the water resistance of the adhesive [20]. Figure 2 displays the effect of different urea concentration on the wet shear strength of the PPI, arachin and conarachin-based adhesives during the combined modification of urea and ECH. Compared with adhesive I0, the wet shear strength of adhesive I2 increased by 53.09%, from 0.81 to 1.24 MPa. With the increase of urea concentration, the globular structure of PPI was gradually unfolded and more active groups were exposed, which made it fully react with epoxy groups to form a dense network structure, thus resulting in advanced performance on water resistance [14,21]. With the addition of urea further increasing, the wet shear strength of adhesive I4 decreased to 0.55 MPa, which was unqualified for the demand of Chinese National Standard GB/T 9846-2015 (greater than or equal to 0.70 MPa). Superfluous urea denatured PPI molecules excessively, which was not conducive to the formation of the network structure, leading to the decline of water resistance [22].

As for the arachin and conarachin-based adhesives, without adding urea, the wet shear strengths of adhesive A0 and C0 were 0.34 and 0.77 MPa, respectively. When the urea concentration was 1 mol l$^{-1}$, the wet shear strengths of adhesive A1 and C1 were maximized to 0.45 and 0.83 MPa respectively, which were 32.35% and 7.79% higher than those without urea treatment. By contrast, the wet shear strengths of adhesive A2 and C2 decreased to 0.19 and 0.46 MPa, respectively, which were decreased by 57.78% and 44.58% compared with their respective maximum values. The reason for the decrease was basically consistent with the PPI-based adhesive. The wet shear strengths of the adhesive A3 and C3 decreased to 0 MPa, reflecting poor water resistance of those adhesives in the case of high urea concentration. Compared with the conarachin component, the change of urea concentration had a more obvious effect on the water resistance of the arachin component during the combined modification of urea and ECH.

## 3.2. Apparent viscosity of the peanut protein isolate, arachin and conarachin-based adhesives

In the process of combined modification of urea and ECH, the effect of different urea concentration on the apparent viscosity of the modified adhesives is indicated in figure 3. Compared with 3.87 Pa s for adhesive I0, the apparent viscosity of adhesive I2 rose to 8.65 Pa s with the addition of urea. The reason was that the molecular structure of PPI was expanded by urea, making it react with ECH more fully to form a certain cross-linked network structure, and the protein cohesion strength increased gradually.

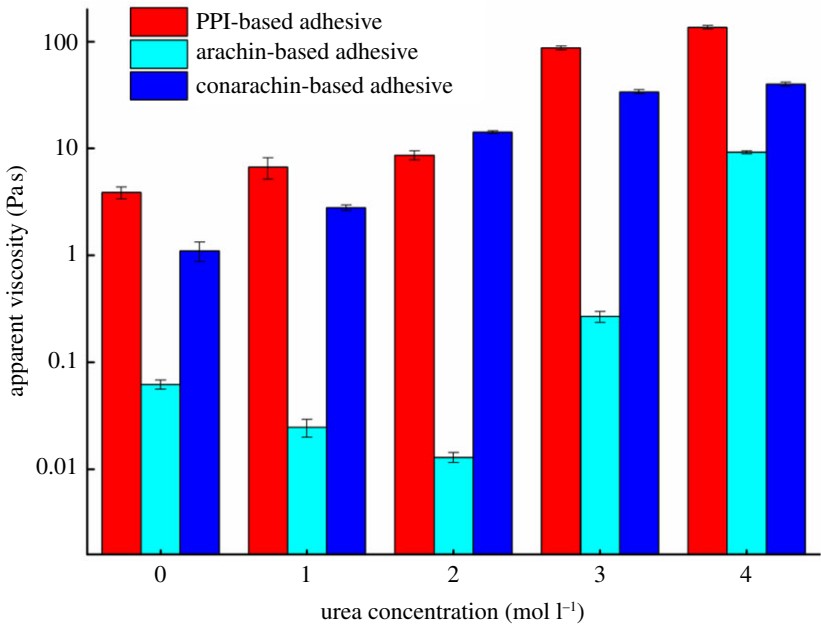

**Figure 3.** Apparent viscosity of the PPI, arachin and conarachin-based adhesives.

A further tremendous increase of the apparent viscosity was observed since the urea concentration reached 3 mol l$^{-1}$. The apparent viscosity of adhesive I3 and I4 increased to 87.43 and 136.80 Pa s, respectively. This behaviour was due to the following reasons: after the treatment with a high concentration of urea, the globular structure of PPI was destroyed into smaller molecular proteins as presented in the SEM results (figure 5). Additionally, the distance between protein molecules became closer and the friction between the protein molecules increased, resulting in the exorbitant apparent viscosity. Excessive viscosity was unfavourable for the shear strength due to the poor fluidity, which was also one of the reasons for the lower wet shear strength of adhesive I3 and I4 [23].

In the case of the arachin-based adhesive, when the urea concentration was 0–3 mol l$^{-1}$, the apparent viscosity of the arachin-based adhesive was low and there was no marked change, while the apparent viscosity of adhesive A4 increased to 9.21 Pa s, which was 14707.07% higher than that without urea. The apparent viscosity of the conarachin-based adhesive showed a stepwise increase with the increase of urea concentration. At urea concentration up to 4 mol l$^{-1}$, the apparent viscosity of adhesive C4 increased from 1.10 to 39.95 Pa s, increasing by 3531.82% more than without urea addition. In the process of the combined modification of urea and ECH, the apparent viscosity of the arachin component was upgraded even more than that of the conarachin component with the increase of urea concentration.

## 3.3. Solid content of the peanut protein isolate, arachin and conarachin-based adhesives

As a basic parameter of plant protein-based adhesives, solid content influences the shear strength of adhesives [24]. Too low solid content leads to fewer gluing components in the adhesive, while too high solid content is unfavourable to uniform sizing because of the high viscosity, both of which will reduce shear strength. As can be seen from figure 4, the solid content of the PPI-based adhesive increased from 18.11% to 31.11% in steps. The amount of urea added per unit volume of the adhesives increased with the increase of urea concentration, leading to a gradual improvement in the solid content.

In accordance with findings on the PPI-based adhesive, the solid contents of the arachin and conarachin-based adhesives were upregulated from 16.91% and 17.87% to 27.12% and 27.78%, respectively, which were 60.38% and 55.46% higher than those without urea treatment. During the combined modification of urea and ECH, the solid content of the arachin component was promoted to a greater extent than that of the conarachin component with the increase of urea concentration.

## 3.4. Scanning electron microscopy analysis

As presented in figure 5, the samples of the PPI, arachin and conarachin-based adhesives were observed at 500× magnification for the protein block size and 5000× magnification for the protein

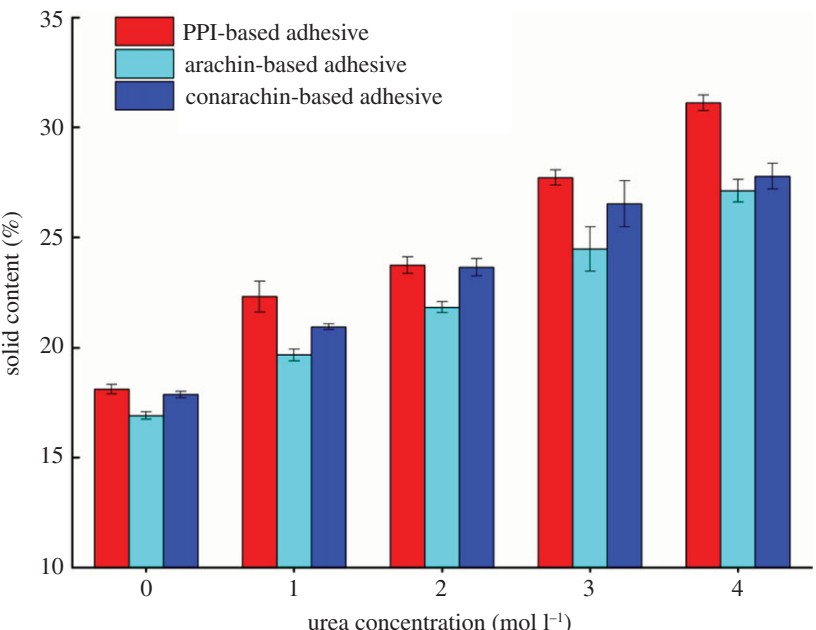

**Figure 4.** Solid content of the PPI, arachin and conarachin-based adhesives.

surface. Without urea treatment, the SEM results of adhesive I0 showed that there were plenty of protein pieces with small size (figure 5, I0-a), and the protein surface was uneven and irregular (figure 5, I0-b). Under this circumstance, the tensile stress in the adhesive layer was easy to form after curing, and the brittleness of the adhesive layer was increased, which was manifested as lower shear strength at the macro level.

Setting against adhesive I0, the protein block size in adhesive I1 and I2 became bigger (figure 5, I1-a and I2-a), which may be due to the formation of the cross-linked network structure in PPI [25]. Furthermore, the protein surface in adhesive I1 and I2 got flatter and denser and the surface cracks and pores were gradually reduced (figure 5, I1-b and I2-b), which can effectively prevent water from intruding into the protein structure and benefit the water resistance.

The protein block size in adhesive I3 and I4 dwindled and the content of protein pieces with small size increased with the further addition of urea (figure 5, I3-a and I4-a). Besides, the size of small block protein in adhesive I4 was more finely broken than that in adhesive I0. Meanwhile, caused by the excessive denaturation of protein due to the high concentration of urea, a few pores were observed in adhesive I3 and I4 (figure 5, I3-b and I4-b). In the water resistance test, water easily penetrated into these pores and cracks, damaging the protein structure, thus resulting in low wet shear strength and poor water resistance of the adhesive [23,26].

In the process of urea concentration increasing from 0 to 1 mol l$^{-1}$, the protein block size in adhesive A1 and C1 enlarged. When the urea concentration continued to increase, the protein block size in the arachin and conarachin-based adhesives decreased, and the content of arachin and conarachin pieces with small size increased gradually. Due to the high concentration of urea, adhesive A4 exhibited irregular protein surface and large cracks appeared on the protein surface in adhesive C4, which was unfit for the improvement of water resistance (figure 5, A4-b and C4-b). The SEM micrographs of arachin and conarachin-based adhesives were basically coincident with the PPI-based adhesive with the change of urea concentration, suggesting that the variations of protein surface and size in the PPI-based adhesive were attributed to both changes of its arachin and conarachin components.

## 3.5. Fourier-transform infrared analysis

The FTIR spectra of the modified adhesives are illustrated in figure 6. As shown in figure 6*a*, the absorption peaks of amide I, II and III bands at 1658, 1528 and 1238 cm$^{-1}$ were ascribed to the stretching vibration of C=O, the in-plane bending of N-H, the stretching vibration of C-N and the bending vibration of N-H, respectively [27,28]. The broad absorption peak at 3000–3500 cm$^{-1}$ was assigned to the stretching vibration of N-H and O-H groups [23].

(*a*)

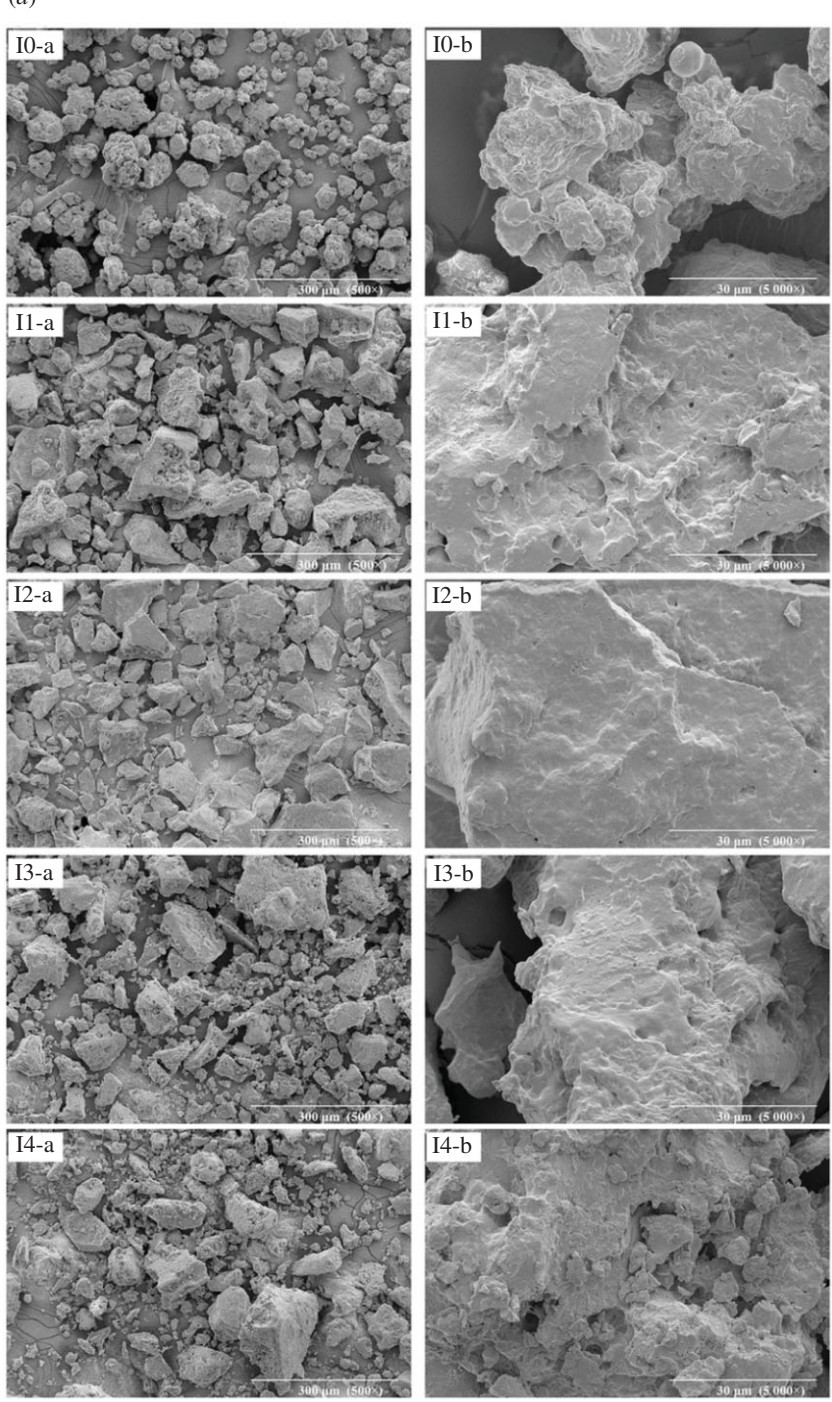

**Figure 5.** SEM images of the PPI, arachin and conarachin-based adhesives (*a*, *b*, *c* represent PPI, arachin and conarachin-based adhesives, 'a' and 'b' represent 500× and 5000× magnification, respectively).

The absorption peaks at 2924 and 2875 cm$^{-1}$ belonged to symmetric and asymmetric -CH$_2$, respectively [29]. The absorption peaks at 1165 and 1740 cm$^{-1}$ arose from the stretching vibration of ether bond and ester bond, respectively [13,30], which were caused by the reaction between hydroxyl and carboxyl group in PPI and epoxy group in ECH to form ether and carboxyl ester structures, promoting the formation of cross-linked network to improve the water resistance of the PPI-based adhesive [31].

The absorption peak ascribed to the bending vibration of COO- was observed at 1385 cm$^{-1}$ in adhesive I0 [32], while the intensity of this corresponding peak in adhesive I1, I2, I3 and I4 weakened.

*(b)*

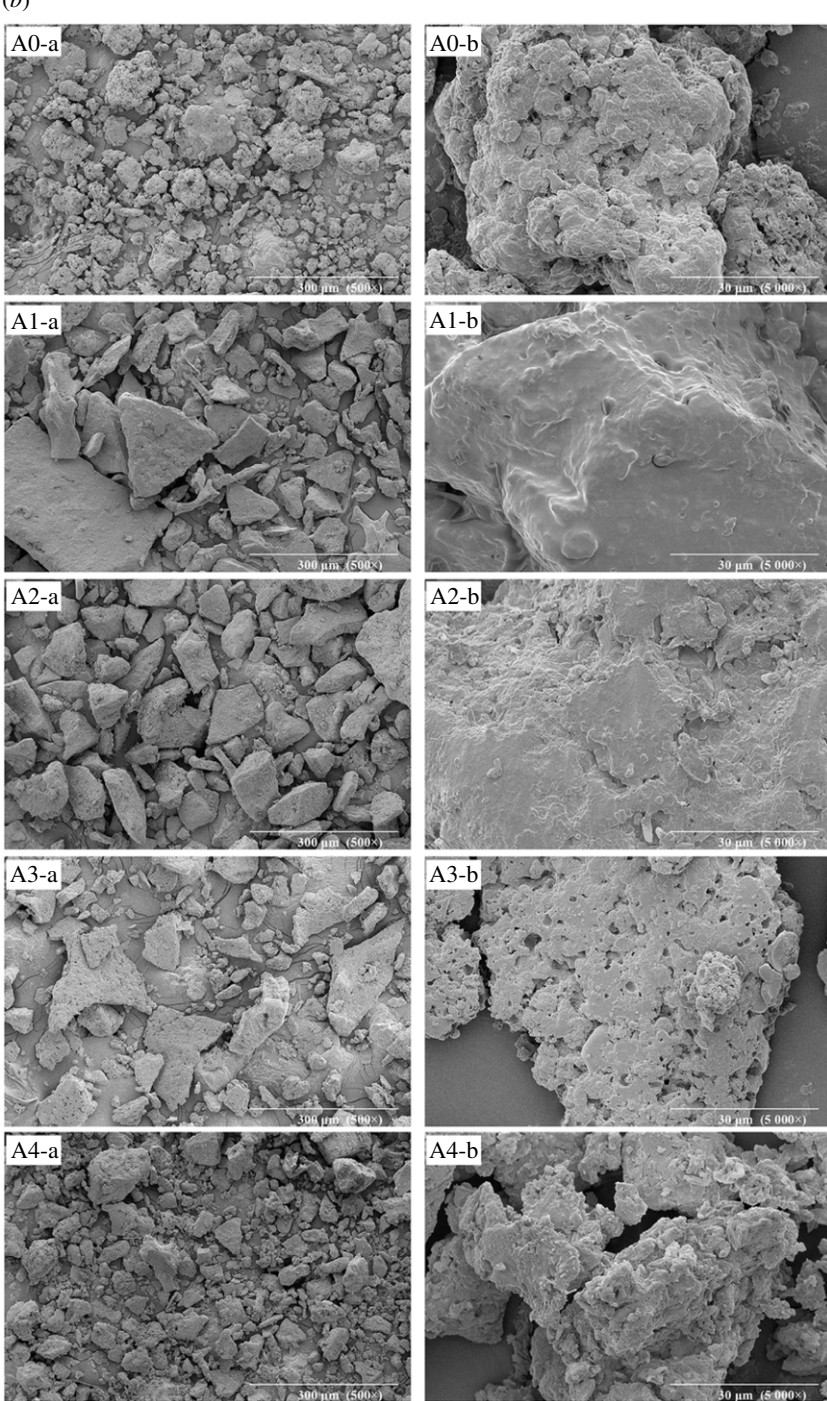

**Figure 5.** *(Continued.)*

Compared with adhesive I1 and I2, the intensity of carboxylate ester absorption peak of adhesive I3 and I4 decreased, indicating that excessive concentration of urea was adverse to the formation of ester bond and cross-linked network structure in PPI-based adhesive.

Except for the intensity of ether bond at 1165 cm$^{-1}$, the position of the main chemical bonds in the arachin-based adhesive (figure 6*b*) were basically the same as those in the PPI-based adhesive (figure 6*a*). As for the conarachin-based adhesive, in addition to the necessary amide I, II and III bands near 1655, 1534 and 1228 cm$^{-1}$, the FTIR results of the conarachin-based adhesive (figure 6*c*) were quite different from those of the PPI-based adhesive and arachin-based adhesive. The differences of the main chemical bonds, especially ether bond and ester bond, may be the reason for the insufficient formation

(c)

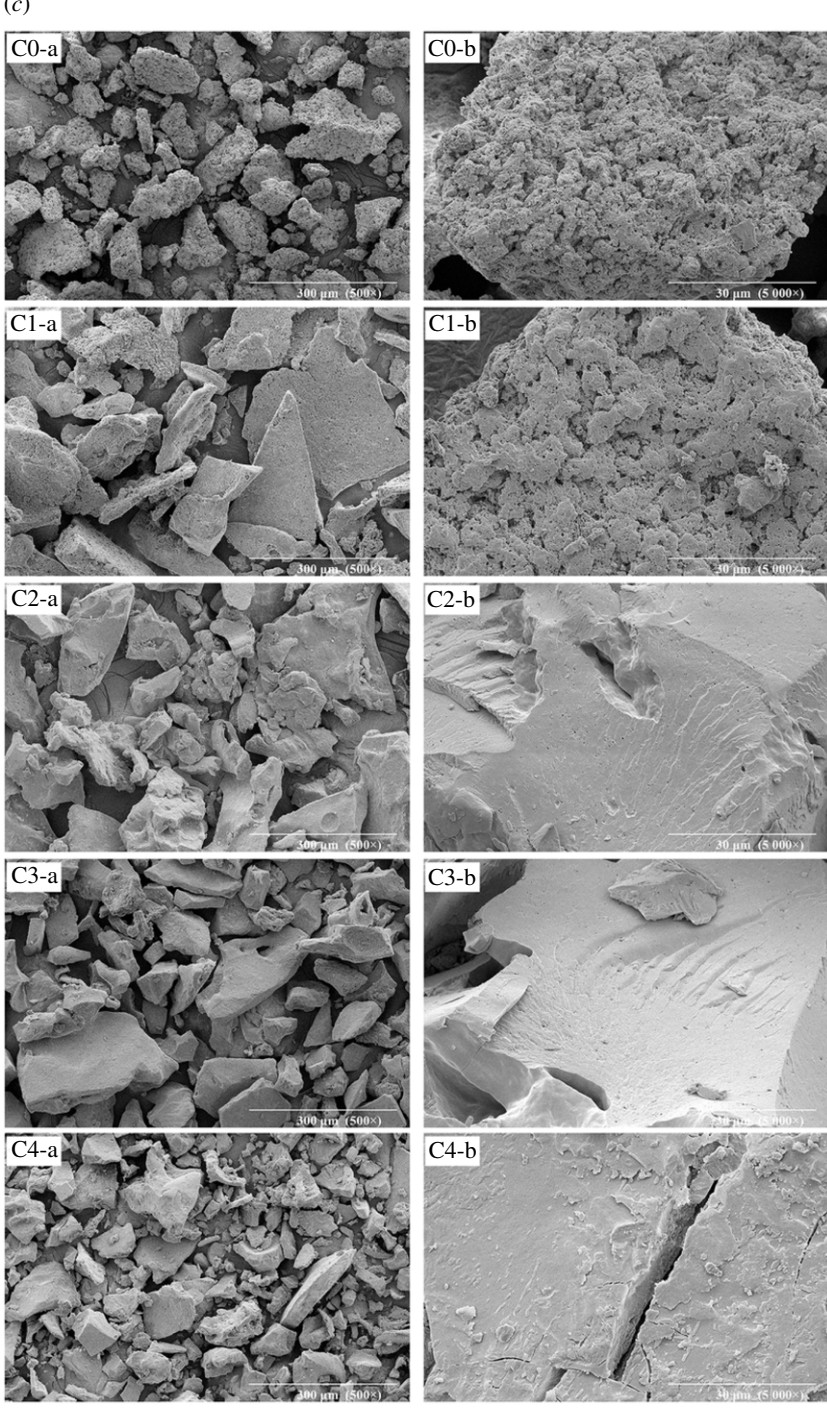

**Figure 5.** (Continued.)

of cross-linking network structure in the arachin and conarachin-based adhesives, resulting in inferior water resistance. The above results indicated that different concentrations of urea treatment caused the changes of the ester bonds, ether bonds and other main chemical bonds in the PPI-based adhesive, which was related to the arachin component.

## 3.6. Thermogravimetry analysis

In the process of combined modification of urea and ECH, the TG and derivative thermogravimetry (DTG) results are depicted in figure 7. As illustrated in figure 7a, the mass loss of the PPI-based

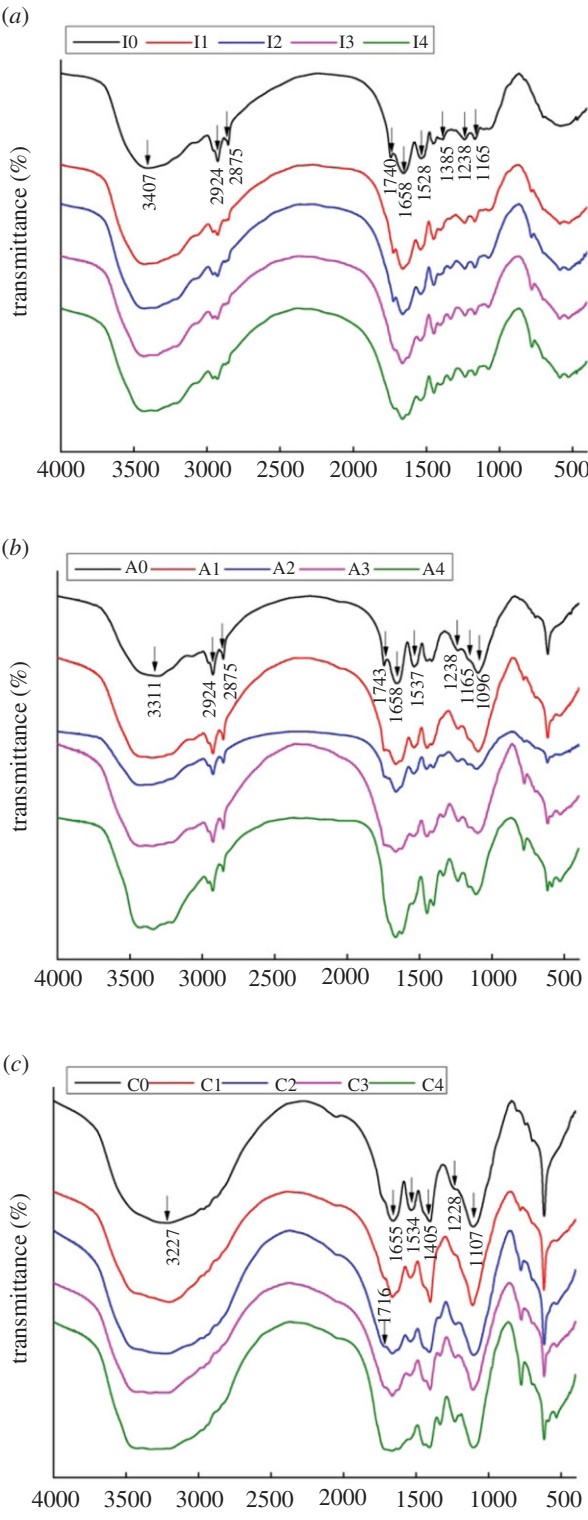

**Figure 6.** FTIR spectra of the PPI, arachin and conarachin-based adhesives.

adhesive could be distinguished by three decline stages. At the first stage (room temperature to 125°C), the mass loss was mainly because of the evaporation of residual moisture in the adhesive [33].

Considered as the initial degradation stage, the second stage was from 125 to 265°C. The mass continued to decline owing to the fracture of some unstable chemical bonds and the decomposition of small molecules [34]. A degradation peak of adhesive I1, I2, I3, I4 was observed at about 192°C, which was ascribed to the degradation of some non-covalent bonds formed after adding urea, such as

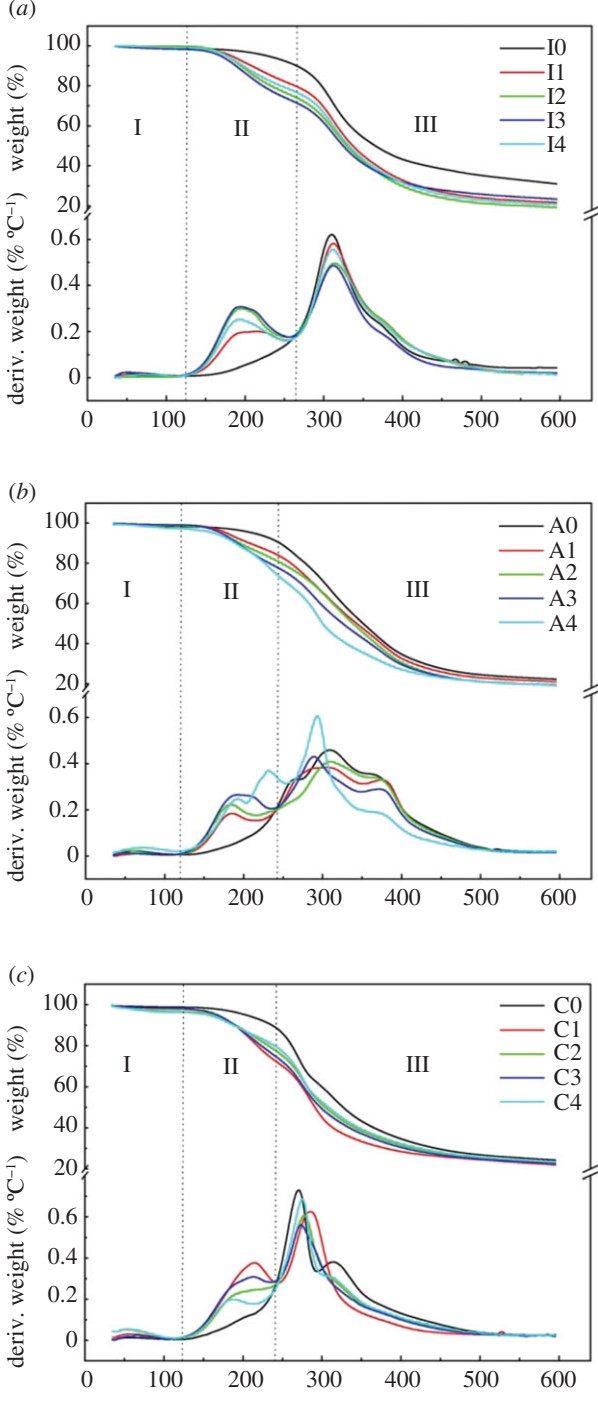

**Figure 7.** TG and DTG curves of the PPI, arachin and conarachin-based adhesives.

electrostatic bonds intramolecular and intermolecular hydrogen bonds [9]. On the other hand, no new degradation peak was observed in adhesive I0 since no urea was added and the degradation rate of the adhesive I0 was slower. Hence, the residual mass of adhesive I0 was higher than that of adhesive I1, I2, I3, I4.

The third stage was from 265 to 600°C. The excessive temperature led to the cleavage of covalent bonds including $C(O)-NH_2$, $C(O)-NH$, C-O, C-N and C-C, and the release of CO, $CO_2$ and $NH_3$ [27], resulting in the decomposition of the main skeleton of protein eventually [35]. As shown in table 1, the temperature corresponding to the fastest loss rate of the adhesive sample was expressed in $T_{max}$, which was assigned to the protein skeleton decomposition [36].

**Table 1.** Degradation peak temperature of the PPI, arachin and conarachin-based adhesives.

| PPI-based adhesive | $T_{max}$ (°C) | arachin-based adhesive | $T_{max}$ (°C) | conarachin-based adhesive | $T_{max}$ (°C) |
|---|---|---|---|---|---|
| I0 | 310.16 | A0 | 309.30 | C0 | 270.50 |
| I1 | 312.16 | A1 | 305.90 | C1 | 285.55 |
| I2 | 314.23 | A2 | 309.85 | C2 | 278.17 |
| I3 | 312.29 | A3 | 289.06 | C3 | 273.57 |
| I4 | 311.22 | A4 | 293.19 | C4 | 275.13 |

As for the adhesive I0, I1, I2, the increase of urea concentration contributed to a higher $T_{max}$, indicating a slight enhancement of the thermal stability of the PPI-based adhesive, which would be in favour of the wet shear strength. Combined with the FTIR results, this may be due to the increase in the number of new chemical bonds formed after adding urea, which made the internal cross-linked network structure of the protein stronger, leading to a sturdy skeleton structure of the protein [36]. The $T_{max}$ of adhesive I3 and I4 decreased to 312.29 and 311.22°C, respectively, which was related to the excessive denaturation of the protein at the exorbitant urea concentration.

The mass loss of arachin and conarachin-based adhesives was divided into three stages as well, and the reasons for mass loss in each stage were the same as that of the PPI-based adhesive. At the third stage of arachin-based adhesive, compared with 309.85°C for adhesive A2, the $T_{max}$ of adhesive A3 was reduced to 289.06°C. Combined with the SEM and FTIR results of arachin-based adhesive, this decline was due to the over-expansion of the arachin structure and the minor arachin block size as well as the lack of new chemical bonds in arachin under high concentration of urea. Compared with 270.50°C for adhesive C0, the $T_{max}$ of adhesive C1 was upregulated to 285.55°C, which was 5.56% higher than that without urea, whereas the $T_{max}$ of adhesive C2 decreased to 278.17°C. The reason for the decrease was similar to that of arachin-based adhesives. The enhancement of the thermal property of conarachin component was improved greatly compared with that of arachin component during the combined modification of urea and ECH.

## 4. Conclusion

During the combined modification of urea and ECH, different urea concentration led to changes in the structure and performance of the PPI-based adhesive, which was attributed to changes in its arachin and conarachin components. As the urea concentration increased from 0 to 4 mol $l^{-1}$, the wet shear strength of the PPI-based adhesive first increased and then decreased, reaching the maximum at the urea concentration of 2 mol $l^{-1}$, while the apparent viscosity and solid content were increased gradually. Compared with conarachin-based adhesives, the change of urea concentration had a more obvious effect on the performance improvement of the arachin-based adhesives during the combined modification.

Based on FTIR results, different urea concentration caused the change of chemical bonds (ester and ether bonds) related to the water resistance enhancement in the PPI-based adhesive, which was mainly associated with the arachin component. TG results suggested that the introduction of urea with appropriate concentration (2 mol $l^{-1}$) brought about increasing levels of thermal behaviour of the PPI-based adhesive. Compared with the arachin component, the thermal stability of the conarachin component was improved greatly. Judged by SEM results, with the increase of urea concentration, the protein block size first increased and then decreased. The adhesive I2 possessed a more compact and denser surface that contributed to the improvement of water resistance, which were mainly related to changes of both arachin and conarachin components.

Data accessibility. Our data are deposited at Dryad Digital Repository: https://doi.org/10.5061/dryad.h70rxwdh8 [37].

Authors' contributions. C.C. and F.C. designed the research. C.C. and Y.D. performed the assay. C.C. and Y.D. analysed the data. C.C. and Y.D. wrote this paper.

Competing interests. We declare we have no competing interests

Funding. Our research was funded by the 13th Five-year National Key Research and Development Plan (grant no. 2018YFD0401100).

Acknowledgements. We really appreciate Yu Liu and Gailing Bao for the help in SEM and FTIR tests, respectively.

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
