## [Peer Review File · Royal Society Open Science]

Review History

RSOS-202227.R0 (Original submission)

Review form: Reviewer 1

Is the manuscript scientifically sound in its present form?

Yes

Are the interpretations and conclusions justified by the results?

Yes

Is the language acceptable?

Yes

Do you have any ethical concerns with this paper?

No

Have you any concerns about statistical analyses in this paper?

Yes

Recommendation?

Accept with minor revision (please list in comments)

Comments to the Author(s)

The article by Chen, Du and Chen discusses the effect of addition of urea on the properties of peanut protein isolate, arachin and conarachin-based adhesives. The authors show that addition of appropriate (2 mol/L) concentrations of urea improve the wet shear strength, solid content, water resistance and so on. This was ascribed to the unfolding of peanut protein isolate as well as exposition of active groups of the peanut protein isolate. This mechanism was corroborated via FTIR measurements. One minor concern about this work is that it is rather closely related to the previous published article cited in reference 13 (in topic and in research content). Here, it would be beneficial if the authors could clearly state the new advances made in this article.

Furthermore, I would like to suggest that the authors show a scheme of the enhancement mechanism. Though the article as is, is readily understandable, a graphic illustrating the mechanism would facilitate understanding of the mechanism.

As the research appears to be interesting and environmental friendly adhesive important for our society, I would like to suggest publication after considering the (mostly minor) comments.

- If possible, can the authors add a scheme denoting the interactions of the adhesives with the substrates (during gluing) and the effect of the urea in this process? Or at least show somewhere the structures of the different glue basis? This would facilitate understanding of the improvement mechanism stated in the article.
- I suggest not to abbreviate urea as it is already a short word, while abbreviating just makes reading of the text more difficult.
- "has more than 8 000 varieties" what kind of varieties? The sentence is too unspecific.
- It is rather difficult to adjust the pH to 9 or 4.5 with stock solutions of 1 M, often leading to overshooting and so on. Why weren't the solutions diluted?
- The experimental section should enable other researchers to repeat the experiments. Therefore, I would like to suggest that the authors add the amount of added salts, such as $(\text{NH}_4)_2\text{SO}_4$ and not use "certain amount". Maybe just give a weight according to an estimated volume.
- What is the difference of the different rows in Figure 5? In the figure captions no information is available on this. It may be advised to add some description in the Figure or the caption. In regard to Figures, Figure 3 may be plotted as log/lin graph (x-axis linear, y-axis log scale)
- I would suggest changing this part of the sentence: "...making it fully react with epoxy groups to form a carboxylate ester structure.". A complete reaction is typically not observed, while a reaction may be shifted more to the products or educts by addition of some chemical. In this regard, it is difficult to conclude this from the given spectra. The authors may supply a magnification of the spectra at that specific wavelength in the supporting information to substantiate their claim. (also advised may be an assisting line)
- In the section about TGA, the authors state that some unstable chemicals are degrading. This is quite unspecific, are these chemicals known?
- Please add concentration in: "introduction of appropriate U (2 mol/L)"
- It may be advised to add a short discussion/comparison of the obtained results with literature values of adhesives? I.e., water resistance, shear strength etc.?
- More recent references may be added to the article (only one reference from 2020).

Review form: Reviewer 2**Is the manuscript scientifically sound in its present form?**

Yes

Are the interpretations and conclusions justified by the results?

Yes

Is the language acceptable?

Yes

Do you have any ethical concerns with this paper?

No

Have you any concerns about statistical analyses in this paper?

No

Recommendation?

Accept with minor revision (please list in comments)

Comments to the Author(s)

1. Why the effect of urea concentration on the properties of peanut protein-based adhesives was studied?
2. Dose the amount of ECH affect the properties of peanut protein-based adhesives?
3. Line 117: What is the protein content of PPI.
4. Line 473, 515: Please check if the format of this citation is correct or not.
5. Line 491: A right parenthesis was missed.

Decision letter (RSOS-202227.R0)

Dear Mr Chen:

Title: Effect of urea concentration on properties of peanut protein isolate, arachin and conarachin-based adhesives during urea-epichlorohydrin modification
Manuscript ID: RSOS-202227

Thank you for submitting the above manuscript to Royal Society Open Science. On behalf of the Editors and the Royal Society of Chemistry, I am pleased to inform you that your manuscript will be accepted for publication in Royal Society Open Science subject to minor revision in accordance with the referee suggestions. Please find the reviewers' comments at the end of this email.

The reviewers and handling editors have recommended publication, but also suggest some minor revisions to your manuscript. Therefore, I invite you to respond to the comments and revise your manuscript.

Because the schedule for publication is very tight, it is a condition of publication that you submit the revised version of your manuscript before 05-Feb-2021. Please note that the revision deadline will expire at 00.00am on this date. If you do not think you will be able to meet this date please let me know immediately.

Kind regards,
Dr Laura Smith
Publishing Editor, Journals

On behalf of the Subject Editor Professor Anthony Stace and the Associate Editor Professor Chaohua Cui.

RSC Associate Editor:
Comments to the Author:
(There are no comments.)

RSC Subject Editor:
Comments to the Author:
(There are no comments.)

Reviewer comments to Author:
Reviewer: 1

Comments to the Author(s)

The article by Chen, Du and Chen discusses the effect of addition of urea on the properties of peanut protein isolate, arachin and conarachin-based adhesives. The authors show that addition of appropriate (2 mol/L) concentrations of urea improve the wet shear strength, solid content, water resistance and so on. This was ascribed to the unfolding of peanut protein isolate as well as exposition of active groups of the peanut protein isolate. This mechanism was corroborated via FTIR measurements. One minor concern about this work is that it is rather closely related to the previous published article cited in reference 13 (in topic and in research content). Here, it would be beneficial if the authors could clearly state the new advances made in this article.

Furthermore, I would like to suggest that the authors show a scheme of the enhancement mechanism. Though the article as is, is readily understandable, a graphic illustrating the mechanism would facilitate understanding of the mechanism.

As the research appears to be interesting and environmental friendly adhesive important for our society, I would like to suggest publication after considering the (mostly minor) comments.

- If possible, can the authors add a scheme denoting the interactions of the adhesives with the substrates (during gluing) and the effect of the urea in this process? Or at least show somewhere the structures of the different glue basis? This would facilitate understanding of the improvement mechanism stated in the article.

- I suggest not to abbreviate urea as it is already a short word, while abbreviating just makes reading of the text more difficult.

- "has more than 8 000 varieties" what kind of varieties? The sentence is too unspecific.

- It is rather difficult to adjust the pH to 9 or 4.5 with stock solutions of 1 M, often leading to overshooting and so on. Why weren't the solutions diluted?

- The experimental section should enable other researchers to repeat the experiments. Therefore, I would like to suggest that the authors add the amount of added salts, such as $(\text{NH}_4)_2\text{SO}_4$ and not use "certain amount". Maybe just give a weight according to an estimated volume.

- What is the difference of the different rows in Figure 5? In the figure captions no information is available on this. It may be advised to add some description in the Figure or the caption. In regard to Figures, Figure 3 may be plotted as log/lin graph (x-axis linear, y-axis log scale)

- I would suggest changing this part of the sentence: "...making it fully react with epoxy groups to form a carboxylate ester structure.". A complete reaction is typically not observed, while a reaction may be shifted more to the products or educts by addition of some chemical. In this regard, it is difficult to conclude this from the given spectra. The authors may supply a magnification of the spectra at that specific wavelength in the supporting information to substantiate their claim. (also advised may be an assisting line)

- In the section about TGA, the authors state that some unstable chemicals are degrading. This is quite unspecific, are these chemicals known?
- Please add concentration in: "introduction of appropriate U (2 mol/L)"
- It may be advised to add a short discussion/comparison of the obtained results with literature values of adhesives? I.e., water resistance, shear strength etc.?
- More recent references may be added to the article (only one reference from 2020).

Reviewer: 2

Comments to the Author(s)

1. Why the effect of urea concentration on the properties of peanut protein-based adhesives was studied?
2. Dose the amount of ECH affect the properties of peanut protein-based adhesives?
3. Line 117: What is the protein content of PPI.
4. Line 473, 515: Please check if the format of this citation is correct or not.
5. Line 491: A right parenthesis was missed.

Author's Response to Decision Letter for (RSOS-202227.R0)

See Appendix A.

RSOS-202227.R1 (Revision)

Review form: Reviewer 1

Is the manuscript scientifically sound in its present form?

Yes

Are the interpretations and conclusions justified by the results?

Yes

Is the language acceptable?

Yes

Do you have any ethical concerns with this paper?

No

Have you any concerns about statistical analyses in this paper?

Yes

Recommendation?

Accept as is

Comments to the Author(s)

The authors have well addressed the comments in the revised manuscript. I am glad to recommend the publication of this work.

Review form: Reviewer 2

Is the manuscript scientifically sound in its present form?

Yes

Are the interpretations and conclusions justified by the results?

Yes

Is the language acceptable?

Yes

Do you have any ethical concerns with this paper?

No

Have you any concerns about statistical analyses in this paper?

No

Recommendation?

Accept as is

Comments to the Author(s)

The revised manuscript is well written and I do not have further comments. I think it is worthy of acceptance.

Decision letter (RSOS-202227.R1)

Dear Mr Chen:

Title: Effect of urea concentration on properties of peanut protein isolate, arachin and conarachin-based adhesives during urea-epichlorohydrin modification

Manuscript ID: RSOS-202227.R1

It is a pleasure to accept your manuscript in its current form for publication in Royal Society Open Science. The chemistry content of Royal Society Open Science is published in collaboration with the Royal Society of Chemistry.

On behalf of the Subject Editor Professor Anthony Stace and the Associate Editor Professor Chaohua Cui.

RSC Associate Editor:
Comments to the Author:
(There are no comments.)

RSC Subject Editor:
Comments to the Author:
(There are no comments.)

Reviewer(s)' Comments to Author:
Reviewer: 2

Comments to the Author(s)
The revised manuscript is well written and I do not have further comments. I think it is worthy of acceptance.

Reviewer: 1

Comments to the Author(s)
The authors have well addressed the comments in the revised manuscript. I am glad to recommend the publication of this work.

Appendix A

Manuscript Draft

Manuscript ID: RSOS-202227

Title: Effect of urea concentration on properties of peanut protein isolate, arachin and conarachin-based adhesives during urea-epichlorohydrin modification

Article Type: Research

Keywords: Peanut protein isolate;
Arachin;
Conarachin;
Wood adhesives;
Urea;
Water resistance

Abstract: To lay theoretical basis for the preparation of peanut protein-based adhesives and promote the sustainable development of adhesive industry, properties of peanut protein isolate (PPI), arachin and conarachin-based adhesives modified by urea and epichlorohydrin (ECH) were investigated under different urea concentrations. When the urea concentration was 2 mol/L, the wet shear strength of the PPI-based adhesive was 1.24 MPa with the best water resistance. With the increase of urea concentration from 0 to 4 mol/L, the apparent viscosity of the PPI-based adhesive increased from 3.87 Pa·s to 136.80 Pa·s and the solid content increased from 18.11% to 31.11%. Compared with conarachin-based adhesive, the properties of arachin-based adhesive were improved more obviously during the combined modification. Scanning electron microscopy images illustrated that when the urea concentration was 2 mol/L, the surface of the PPI-based adhesive was more compact and smoother, which was beneficial to the improvement of water resistance and related to the structure changes of arachin and conarachin components. Fourier-transform infrared spectroscopy results indicated that different urea concentrations caused the change of ester and ether bonds in the PPI-based adhesive, which was mainly related to arachin component. Thermogravimetry results suggested that when the urea concentration was 2 mol/L, the decomposition temperature of protein skeleton in the PPI-based adhesive reached a maximum of 314 °C exhibiting the highest thermal stability. The improvement of the thermal stability of conarachin was greater than that of arachin during the combined modification.

<Journal Name>: Royal Society Open Science

< Manuscript ID>: RSOS-202227

<Manuscript Title>: Effect of urea concentration on properties of peanut protein isolate, arachin and conarachin-based adhesives during urea-epichlorohydrin modification

Dear Editors:

Thank you very much for your kind information. We highly appreciate the meaningful comments from editor and reviewers concerning our manuscript entitled **“Effect of urea concentration on properties of peanut protein isolate, arachin and conarachin-based adhesives during urea-epichlorohydrin modification”**. Those comments are all valuable and very helpful for revising and improving our paper, as well as the important guiding significance to our researches. We have carefully considered the comments and have tried our best to revise the manuscript according to the comments.

Enclosed please find the revised manuscript, which we would like to submit for your kind consideration. A point-by-point list of our response (highlighted in blue) to the Reviewers’ and Editor’s comments have been attached.

We hope this manuscript is suitable for **“Royal Society Open Science”**. We would like to express our great appreciation to you and reviewers for comments on our paper. Looking forward to hearing from you.

Thank you and best regards,

Sincerely yours,

Fusheng Chen, Ph.D.

Professor

Henan University of Technology

Zhengzhou, 450001, P.R. China

E-mail: fushengc@haut.edu.cn

Incl.: Responses (in blue) to the comments from reviewers.

Response to Referees

Reviewer: 1

Comments to the Author(s)

The article by Chen, Du and Chen discusses the effect of addition of urea on the properties of peanut protein isolate, arachin and conarachin-based adhesives. The authors show that addition of appropriate (2 mol/L) concentrations of urea improve the wet shear strength, solid content, water resistance and so on. This was ascribed to the unfolding of peanut protein isolate as well as exposition of active groups of the peanut protein isolate. This mechanism was corroborated via FTIR measurements. One minor concern about this work is that it is rather closely related to the previous published article cited in reference 13 (in topic and in research content). Here, it would be beneficial if the authors could clearly state the new advances made in this article. Furthermore, I would like to suggest that the authors show a scheme of the enhancement mechanism. Though the article as is, is readily understandable, a graphic illustrating the mechanism would facilitate understanding of the mechanism. As the research appears to be interesting and environmental friendly adhesive important for our society, I would like to suggest publication after considering the (mostly minor) comments.

Authors' response: Thanks for reviewer's insightful comment. Firstly, in "reference 13", the protein content of the raw material (peanut meal) was 50.12% containing a certain amount of non-protein components, such as polysaccharide, while the peanut protein isolate (PPI) with protein content of 90.68% was used as raw material in this research, and the interference of other non-protein components on the results was excluded as much as possible.

Secondly, "reference 13" only studied the case of urea concentration of 1 mol/L, but this research studied the performance of the adhesives modified by different urea concentrations (0, 1, 2, 3, 4 mol/L) on the basis of "reference 13", which had a wider exploration scope and could provide more theoretical basis for practical applications. From the results, the wet shear strength of the PPI-based adhesive was 1.24 MPa with the best water resistance, which was higher than that of peanut meal-based adhesive (0.92 MPa).

Thirdly, compared with "reference 13", the performance of arachin and conarachin, the main components of peanut protein, were also studied under different urea concentrations in this research. In general, this research is a further study based on "reference 13".

Moreover, according to the reviewer's insightful comment, a graphic illustrating the mechanism have been made by us to facilitate understanding of the mechanism. Please refer to the Figure 1 below.

Figure 1 Graphical abstract

Question 1: If possible, can the authors add a scheme denoting the interactions of the adhesives with the substrates (during gluing) and the effect of the urea in this process? Or at least show somewhere the structures of the different glue basis? This would facilitate understanding of the improvement mechanism stated in the article.

Authors' response: Thanks for reviewer's useful comment. The scheme denoting the interactions of the adhesives with the substrates (during gluing) was presented in Figure 2 below and the effect of the urea in this process please refer to Figure 1 above.

Figure 2 The interactions of the adhesives with the substrates during gluing

Question 2: I suggest not to abbreviate urea as it is already a short word, while abbreviating just makes reading of the text more difficult.

Authors' response: Thanks for reviewer's useful comment. We are sorry about the inappropriate abbreviation. We have replaced "U" with "urea" in the full manuscript. Please refer to the revised edition.

Question 3: "has more than 8 000 varieties" what kind of varieties? The sentence is too unspecific.

Authors' response: Thanks for reviewer's careful comment. We are sorry about the unspecific expression. We have deleted the "has more than 8 000 varieties" and rewritten this sentence. Please refer to line 69 in the revised edition.

Question 4: It is rather difficult to adjust the pH to 9 or 4.5 with stock solutions of 1 M, often leading to overshooting and so on. Why weren't the solutions diluted?

Authors' response: Thanks for reviewer's insightful comment. During the actual preparation of PPI, overshooting phenomenon may occur. Therefore, when the pH is

about to reach 9 or 4.5, we will dilute the stock solutions to 0.5 M according to the actual situation to avoid overshooting.

Question 5: The experimental section should enable other researchers to repeat the experiments. Therefore, I would like to suggest that the authors add the amount of added salts, such as $(\text{NH}_4)_2\text{SO}_4$ and not use “certain amount”. Maybe just give a weight according to an estimated volume.

Authors’ response: Thanks for reviewer’s insightful comment. Indeed, the experimental section should enable other researchers to repeat the experiments. In order to achieve the required saturation, the amount of $(\text{NH}_4)_2\text{SO}_4$ was added according to the supernatant volume and saturation each time. Specific addition amount of $(\text{NH}_4)_2\text{SO}_4$ was presented in the “Electronic supplementary materials”. Please refer to line 124-125 in the revised edition and the Table S1 in “Electronic supplementary materials”.

Question 6: What is the difference of the different rows in Figure 5? In the figure captions no information is available on this. It may be advised to add some description in the Figure or the caption. In regard to Figures, Figure 3 may be plotted as log/lin graph (x-axis linear, y-axis log scale).

Authors’ response: Thanks for reviewer’s careful comment. Different rows represent treatments with different concentrations of urea. We are sorry for the unclear Figure 5. We have revised the Figure 5 and added some description to make it clearly visible. Moreover, the Figure 3 has been plotted as log/lin graph (x-axis linear, y-axis log scale) according to the reviewer’s insightful comment. Please refer to line 534-536, Figure 3 and Figure 5 in the revised edition.

Question 7: I would suggest changing this part of the sentence: “...making it fully react with epoxy groups to form a carboxylate ester structure.”. A complete reaction is typically not observed, while a reaction may be shifted more to the products or educts by addition of some chemical. In this regard, it is difficult to conclude this from the given spectra. The authors may supply a magnification of the spectra at that specific wavelength in the supporting information to substantiate their claim. (also advised may be an assisting line)

Authors’ response: Thanks for reviewer’s insightful comment. We have deleted the “The reason was that...making it fully react with epoxy groups to form a carboxylate ester structure.” Please refer to line 324 in the revised edition.

Question 8: In the section about TGA, the authors state that some unstable chemicals are degrading. This is quite unspecific, are these chemicals known?

Authors’ response: Thanks for reviewer’s useful comment. According to Liu et al., at the second stage, some unstable chemical bonds, such as intermolecular and intramolecular hydrogen bonds and electrostatic bonds, and hydrophobic interaction decomposed [1].

Question 9: Please add concentration in: “introduction of appropriate U (2 mol/L)”

Authors’ response: Thanks for reviewer’s careful comment. We have changed “appropriate U (2 mol/L)” to “urea with appropriate concentration (2 mol/L)”. Please refer to line 398 in the revised edition.

Question 10: It may be advised to add a short discussion/comparison of the obtained results with literature values of adhesives? I.e., water resistance, shear strength etc.?

Authors’ response: Thanks for reviewer’s useful comment. As shown in Table 1 below, the wet shear strength obtained in our research was compared with the reported literatures.

In our research, under the urea concentration of 2 mol/L, the wet shear strength of the PPI-based adhesive was 1.24 MPa with the best water resistance, which met the requirements of Chinese National Standard (0.7 MPa).

The different protein isolates with protein contents of more than 90% were used as the raw materials in literature 1, 2 and 3. The wet shear strength of the PPI-based adhesive in our research was 22.77% and 8.77% higher than that in literature 1 and 2, but lower than that in literature 3. The difference in wet shear strength may be caused by different modification methods and conditions, leading to the different water resistance.

The raw materials used in literature 4 and 5 were soybean flour with low protein content. The wet shear strength of the PPI-based adhesive in our research was 26.53% and 45.88% higher than that in literature 4 and 5. On the one hand, the difference in wet shear strength may be due to different modification methods, on the other hand, it may be due to the low protein content of their raw materials.

Table 1 Properties comparison of the adhesives with different modification treatments

References	Materials	Modification conditions	Best wet shear strength (MPa)
This research	Peanut protein isolate	Urea, Epichlorohydrin	1.24
Literature 1 [2]	Sesame protein isolate	Urea, Zinc oxide	1.01
Literature 2 [3]	Soybean protein isolate	Waterborne epoxy emulsion	1.14
Literature 3 [4]	Soybean protein isolate	Aramid fibers	1.68
Literature 4 [5]	Soybean flour	Silane coupling agent	0.98
Literature 5 [6]	Soybean flour	Glyoxal	0.85

Question 11: More recent references may be added to the article (only one reference from 2020).

Authors’ response: Thanks for reviewer’s careful comment. We have added more recent references. Please refer to line 51, 58, 60, 210, 430-432, 441-445, 480-482 in the revised edition.

Reviewer: 2

Comments to the Author(s)

1. Why the effect of urea concentration on the properties of peanut protein-based adhesives was studied?

Authors' response: Thanks for reviewer's useful comment. In our previous research, we found that the properties of peanut meal-based adhesive were enhanced by the combined modification of urea solution (1 mol/L) and epichlorohydrin (ECH). Urea is widely used in the structural destruction of proteins with its unique properties. As a protein denaturant, urea could unfold the structure of proteins expose the inner active groups such as $-NH_2$, $-COOH$ and $-OH$ [7]. We speculate that different concentrations of urea treatment may lead to different changes in protein structure, resulting in different adhesive properties. Therefore, based on our previous research, the performance changes of peanut protein-based adhesives under different urea concentrations (0, 1, 2, 3, 4 mol/L) were further explored.

2. Dose the amount of ECH affect the properties of peanut protein-based adhesives?

Authors' response: Thanks for reviewer's precious comment. ECH can react with the reactive groups of the proteins via its epoxy group and generate new firm chemical bonds, which make the interior of the protein structure more stable [8]. The amount of ECH will affect the number of epoxy groups that react with urea in the adhesive system, leading to the performance variation possibly. This comment provides a considerable reference for us to investigate the effects of ECH addition on the properties of peanut protein-based adhesives in our future research.

3. Line 117: What is the protein content of PPI.

Authors' response: Thanks for reviewer's useful comment. The protein content of PPI was 90.68%, which was determined using the Kjeldahl method.

4. Line 473, 515: Please check if the format of this citation is correct or not.

Authors' response: Thanks for reviewer's careful comment. We are sorry about the spelling mistake. We have corrected the format of these citations. Please refer to reference [19], [35] in line 475 and line 520 in the revised edition.

5. Line 491: A right parenthesis was missed.

Authors' response: Thanks for reviewer's careful comment. We are sorry about the mistake. We have corrected the format of the citation. Please refer to reference [26] in line 496 in the revised edition.

References

1. Liu H, Li C, Sun XS. 2015 Improved water resistance in undecylenic acid (UA)-modified soy protein isolate (SPI)-based adhesives. *Ind. Crops Prod.* **74**, 577-584. (doi:10.1016/j.indcrop.2015.05.043)
2. Wei X, Wang X, Li Y, Ma Y. 2017 Properties of a new renewable sesame protein adhesive modified by urea in the absence and presence of zinc oxide. *Rsc Adv.* **7**, 46388-46394. (doi: 10.1039/C7RA07578B)
3. Pang H, Zhao S, Wang Z, Zhang W, Zhang S, Li J. 2020 Development of soy protein-based adhesive with high water resistance and bonding strength by waterborne epoxy crosslinking strategy. *Int. J. Adhes. Adhes.* **100**, 102600. (doi: 10.1016/j.ijadhadh.2020.102600)
4. Gu W, Liu X, Ye Q, Gao Q, Gong S, Li J, Shi SQ. 2020 Bio-inspired co-deposition strategy of aramid fibers to improve performance of soy protein isolate-based adhesive. *Ind. Crops Prod.* **150**, 112424. (doi: 10.1016/j.indcrop.2020.112424)
5. Li C, Li H, Zhang S, Li J. 2014 Preparation of reinforced soy protein adhesive using silane coupling agent as an enhancer. *Bioresources.* **9**, 5448-5460. (doi:10.15376/biores.9.3.5448-5460)
6. Wu Z, Liang J, Lei H, Zhang B, Li L. 2021 Study on the soy protein-based adhesive cross-linked by glyoxal. *J. Renew. Mater.* **9**, 205-218. (doi:10.32604/jrm.2021.013655)
7. Sun X, Bian K. 1999 Shear strength and water resistance of modified soy protein adhesives. *J. Am. Oil Chem. Soc.* **76**, 977-980. (doi: 10.1007/s11746-999-0115-2)
8. Mousavi SY, Huang J, Li K. 2018 Investigation of poly (glycidyl methacrylate-co-styrene) as a curing agent for soy-based wood adhesives. *Int. J. Adhes. Adhes.* **82**, 67-71. (doi: 10.1016/j.ijadhadh.2017.12.017)